



# Satellite-derived sea-ice export and its impact on Arctic ice mass balance

Robert Ricker[1,2], Fanny Girard-Ardhuin[2], Thomas Krumpen[1], and Camille Lique[2]

[1]Alfred Wegener Institute, Helmholtz Centre for Polar and Marine Research, Bremerhaven, Bussestrasse 24, 27570 Bremerhaven, Germany

[2]Univ. Brest, CNRS, IRD, Ifremer, Laboratoire d'Oceanographie Physique et Spatiale (LOPS), IUEM, 29280, Brest, France

*Correspondence to:* Robert Ricker (Robert.Ricker@awi.de)

**Abstract.** Ice volume export drives variations of Arctic ice mass balance. It also represents a significant fresh water input to the North Atlantic, which could in turn modulate the intensity of the thermohaline circulation. We present the first estimates of winter sea ice volume export through the Fram Strait using CryoSat-2 sea ice thickness retrievals and three different drift products for the years 2010 to 2017. The export rates vary between -21 and -540 km³/month. We find that ice drift variability

is the main driver of annual and interannual ice volume export variability, and that the interannual variations of the ice drift are driven by large scale variability of the atmospheric circulation captured by the Arctic Oscillation and North Atlantic Oscillation indices. On shorter timescale, however, the seasonal cycle is also driven by the mean thickness of exported sea ice, typically peaking in March. Considering Arctic winter multiyear ice volume changes, 54 % of the variability can be explained by the variations of ice volume export through the Fram Strait.

# 1  Introduction

Variability of the Arctic sea ice export contributes significantly to the variations of surface salinity in the subpolar gyre, and in particular in the regions where deep convection occurs, such as the Labrador and Greenland Seas. Fram Strait ice export represents approximately 25% of the total fresh water export to the North Atlantic (Lique et al., 2009). By the impact on convective overturning of water masses in the North Atlantic, changes in the export rates could affect the global ocean

thermohaline circulation (Dickson et al., 1988). A recent study by Ionita et al. (2016) reports that persistent atmospheric blocking in winter leads to increased sea ice export through the Fram Strait, causing abrupt shifts in the Atlantic meridional overturning circulation variability. In turn, this might also affect the climate over Europe.

Variability of the Arctic sea-ice mass balance is determined by sea-ice production and melt on the one hand, and sea ice export on the other hand. Arctic sea ice volume and related interannual variations have been investigated in various recent

studies (Tilling et al., 2015; Kwok and Cunningham, 2015; Ricker et al., 2017a). The Fram Strait represents the main Arctic gate for sea ice export. While ice export rates during summer are relatively low (Krumpen et al., 2016), winter ice export plays an important role for the multiyear ice (MYI) mass balance in the Arctic (Kwok et al., 1999). MYI is defined as sea ice that survived at least one summer melt period and significantly contributes to the exported sea ice (Kwok et al., 1999). Its greater age implies that it went through a longer period of thermodynamic ice growth and additional thickening due to deformation.





Therefore, MYI can reach several meters of thickness, making it resistant against storms and melting. Hence, MYI attenuates potential loss of ice coverage due to external forcing, while the thinner first-year ice (FYI) is much more sensitive to storms and temperature fluctuations (Holland et al., 2006). As a consequence, summer ice concentration strongly correlates with MYI coverage, highlighting its climate relevance (Comiso, 1990; Thomas and Rothrock, 1993). Multiple studies have shown that

Arctic MYI fraction has been shrinking during the last decades, from about 75% in the mid 1980s to 45% in 2011 (Maslanik et al., 2011). Indeed, anomalously large summer melt reduces the MYI volume and prevents its replenishment by aging FYI (Stroeve et al., 2014; Kwok, 2007).

In order to improve our understanding of these processes that are linked to the variability of Arctic MYI mass balance, monitoring sea ice volume export through the Fram Strait is crucial. Only satellite measurements have the capability to continuously

monitor pan-arctic changes in ice concentration, thickness and drift, the parameters required for calculating ice volume flux. Spreen et al. (2009) estimated Fram Strait sea ice volume export between 2003 and 2008. They used ICESat laser altimeter observations to derive sea ice thickness and AMSR-E 89 GHz passive microwave data to retrieve sea ice concentration and drift. A comparison with previous estimates that were based on a parametrization of ice thickness (Vinje et al., 1998; Kwok and Rothrock, 1999) and drift (Vinje et al., 1998) did not indicate a significant change of the total amount of Fram Strait sea

ice export between the 1990s and 2008. However, one needs to keep in mind that ICESat measurements were restricted to two periods per winter season, and thus, investigations on the seasonal cycle of ice volume export were limited. The European Space Agency (ESA) satellite CryoSat-2 (CS2) was launched in 2010 and partly overcomes these limitations (Wingham et al., 2006) as monthly Arctic wide CS2 sea-ice thickness estimates are derived between October and April (Tilling et al., 2016; Ricker et al., 2014).

In this study, we pursue four main objectives. First, we use the CS2 ice thickness data set (Ricker et al., 2014) to estimate for the first time winter sea ice export through Fram Strait over 7 years between 2010 and 2017 and compare our estimates with previous studies. We use three different low-resolution ice drift products in order to access the impact of the chosen drift data set. Second, we aim to examine the temporal variability of volume export and its links with variability in sea ice drift, thickness and concentration. We then relate the interannual variability of ice volume export through Fram Strait to the variability of the

atmospheric circulation captured by the Arctic Oscillation and North Atlantic Oscillation indices. Our fourth objective is to quantify the impact of winter ice volume export on Arctic sea ice mass balance, which will be achieved by considering Arctic net monthly ice volume changes.

The paper is organized as follows. Section 2 describes the CS2 ice thickness product, the used ice drift data and ancillary data sets. In section 3, we first examine spatial and temporal variability of sea ice thickness, drift and ice concentration at the

Fram Strait gate and present estimates of the ice volume flux and Fram Strait export. The seasonal and interannual variability of ice volume export and its impact on Arctic ice mass balance are discussed in section 4. Conclusions are drawn in section 5.





**Table 1.** Ice drift products used for this study.

| Product | Input data | Temporal resolution | Spatial resolution | Period |
|---------|-----------|---------------------|--------------------|--------|
| OSI-405 (merged) | SSMIS (91 GHz, DMSP F17), ASCAT (Metop-B), AMSR-2 (18.7 and 36.5 GHz) | 48 h | 62.5 km | 2009 - 2017 |
| CERSAT (merged) | QuikSCAT, ASCAT (MetOp-A , Metop-B), SSM/I (85 GHz), SSMIS (91 GHz) | 1 month | 62.5 km | 1991 - 2017 |
| NSIDC | AMSR-E (89 GHz), SSM/I (85 GHz), SMMR (37 GHz), AVHRR, buoy position, NCEP/NCAR wind data | 1 month | 25 km | 1978 - 2015 |

## 2 Data and Methods

In this section, we describe data products used in this study, as well as methods to retrieve ice volume fluxes through the Fram Strait. Table 1 summarizes the specifications of the ice drift products. In addition to drift, ice thickness and concentration data are required to estimate ice volume fluxes.

### 2.1 Sea Ice drift

#### 2.1.1 OSI SAF

We use the low resolution sea ice drift data set from the Ocean and Sea Ice Satellite Application Facility (OSI SAF), specifically the OSI-405 multi sensor product. Various sensors and channels are processed in order to produce the merged product used here: SSMIS (91 GHz H&V polarization) on board DMSP platform F17, ASCAT (C-band backscatter) on board platform Metop-A, and AMSR-2 on board JAXA platform GCOM-W. Ice drift is estimated by an advanced cross-correlation method (Continuous MCC) on pairs of satellite images (Lavergne et al., 2010). The merged product considers the different single sensor data and their quality statistics in order to compensate for data gaps in the single sensor products. We use this multi sensor data set, since we require a sufficient data coverage in the Fram Strait area, which is not given by the single sensor products. Displacements and geographic coordinates of the start and end point of the displacements for 48 h time spans are provided on a 62.5 km x 62.5 km polar stereographic grid. In the following we refer to this product as *OSISAF*.

#### 2.1.2 Ifremer

From Ifremer/CERSAT, we use the merged product, which is obtained from combining Advanced Scatterometer (ASCAT) data and special sensor microwave/imager (SSM/I) brightness temperature measurements. It is provided for different time spans, including monthly lags, which is suitable for our study. The algorithm to deduce ice drift from scatterometer data and the merging with radiometer data is described in Ezraty et al. (2007) and Girard-Ardhuin and Ezraty (2012). Geographic



coordinates of the start and end point of the displacements are provided on a 62.5 km x 62.5 km polar stereographic grid. In the following we refer to this product as *IFREMER*.

### 2.1.3 NSIDC

Finally, we also use the Polar Pathfinder Sea Ice Motion Vectors (version 2), distributed by the National Snow and Ice Data Center (NSIDC). It provides a year-round ice drift data set. As for OSISAF and IFREMER, ice drift is obtained from multiple satellite sensors including radiometers and scatterometers (Table 1) complemented by buoy observations from the International Arctic Buoy Program (IABP). During summer, NCEP/NCAR winds speeds are used to estimate ice drift when satellite data are not available. Though we do not make use of the summer ice drift data, we choose to include this data set, since it is widely used in other studies (e.g. Krumpen et al. (2016) and Spreen et al. (2011)). Monthly Displacements in x and y direction are provided on an EASE 2 25 km x 25 km polar stereographic grid. In the following we refer to this product as *NSIDC*.

### 2.2 AWI CS2 sea ice thickness

We use the AWI CS2 product (processor version 1.2). Processing is based on CS2 orbit data files provided by ESA. Radar waveforms are processed according to Hendricks et al. (2016) and Ricker et al. (2014), using a 50% threshold-first-maximum retracker to obtain ellipsoidal surface elevations (Ricker et al., 2014; Helm et al., 2014). Radar waveforms from surfaces that contain openings in the ice pack appear as specular echoes and can be separated from diffuse echoes that contain reflections from sea ice only. Based on this surface type classification, open water elevations are identified and used to derive the instantaneous sea-surface height anomaly by interpolation. To retrieve sea ice freeboard, the sea-surface height anomaly is subtracted from the ice surface elevations.

Freeboard is converted into sea ice thickness by assuming hydrostatic equilibrium (Laxon et al., 2003). For the conversion, we use ice densities of 916.7 kg/m$^3$ and 882.0 kg/m$^3$ for FYI and MYI respectively (Alexandrov et al., 2010), and 1024 kg/m$^3$ for the sea water density. Snow depth and density are deduced from the Warren snow climatology (W99) (Warren et al., 1999). The climatology is modified by reducing the snow depth by 50 % over FYI to take into account the recent change towards a seasonal Arctic ice cover. FYI and MYI are identified with the daily OSI SAF sea ice type product (Eastwood, 2012). In order to obtain a sufficient spatial coverage, acquired thickness data are averaged monthly on an 25 km EASE 2 grid.

The CS2 observational uncertainties of sea ice thickness contain contributions that are associated with speckle noise, sea-surface height estimation and densities of ice and snow (Ricker et al., 2014). They can easily reach values of > 1 m for single measurements, but will be reduced to the range of centimeters by spatial averaging. Note that during the melting period from May to September, the presence of melt ponds prevents the retrieval of sea ice thickness observations.

### 2.3 OSI SAF Ice concentration and type

We use the OSI SAF sea-ice concentration and type products (OSI-401-b) and (OSI-403-b), respectively (Eastwood, 2012). Ice concentration is needed for the ice volume computation for each 25 km grid cell and ice type is used to classify grid cells





as FYI or MYI. The products are updated daily and the data are provided on a 10 km polar stereographic grid. To be consistent with the CS2 product, monthly means are projected onto the EASE2 25 km grid. Grid cells originally flagged as *ambiguous* are replaced by an inverse-distance interpolation to obtain FYI or MYI flags for all ice-covered grid cells.

### 2.4 Retrieving ice volume flux and export rates through Fram Strait

The first step is to project the ice drift and thickness data onto a common grid. The EASE 2 grid is based on an equal area projection, and therefore, it is reasonable to use it for sea-ice volume estimations (Ricker et al., 2017a). Hence, we define the 25 km EASE 2 grid provided in the AWI CS2 ice thickness product as our standard grid and interpolate the displacement data onto this grid. Since the NSIDC displacement data are already projected on an EASE grid, we only interpolate the displacements in x and y direction onto the 25 km grid. In contrast, the IFREMER and OSISAF grids are based on a polar stereographic

projection. Here, we use the geographic coordinates of the start and end point of the displacement and project them onto the EASE 2 grid separately. Afterwards, displacements in x and y direction of the EASE 2 grid are calculated. Since the Ifremer product is provided as monthly means, the daily updated OSISAF 48 h displacements need to be averaged monthly. Here, we calculate the displacements in x and y direction on the EASE 2 grid for each day and average them over one month.

Monthly ice volume flux $Q_{x,y}$ in x and y direction is obtained by:

$$Q_{xy} = lHCD_{xy}, \tag{1}$$

where $l = 25$ km is the size of the grid cells, H is the CS2 sea ice thickness, C is the ice concentration obtained from the OSISAF product, and $D_{xy}$ represents the ice drift in x and y direction respectively.

In order to compute ice volume export through Fram Strait, we follow the methodology of Krumpen et al. (2016) and define a gate that is a composite of a meridional and a zonal gate (Figure 1). The meridional gate is located along 82°N between

12°W and 20°E. The zonal part is located along 20°E between 80.5°N and 82°N. We have chosen this gate location to reduce errors and biases in low resolution ice drift data that become larger with increasing ice velocities, typically found south of 82°N (Sumata et al., 2014, 2015). Moreover, uncertainty of CS2 ice thickness increases at lower latitudes, especially near Fram Strait due to sparse orbit coverage (Ricker et al., 2014).

Meridional ($Q_v$) and zonal ($Q_u$) components of the ice volume flux through the defined gate are calculated as follows:

$$Q_v = l/\cos(\lambda)HC(D_x\sin(\lambda) - D_y\cos(\lambda))$$
$$Q_u = l/\cos(\lambda)HC(D_x\cos(\lambda) - D_y\sin(\lambda)), \tag{2}$$

where $\lambda$ is the longitude of the respective grid cell. Uncertainties of $Q$ are estimated by:

$$\sigma_Q = l/\cos(\lambda)\sqrt{(HC\sigma_D)^2 + (DC\sigma_H)^2 + (HD\sigma_C)^2}. \tag{3}$$

Consistent with Laxon et al. (2013) and Ricker et al. (2017a), we set the ice-concentration uncertainty $\sigma_{c_i} = 5\%$. Neverthe-

less, we acknowledge that the uncertainty may vary depending on the actual ice concentration (Ivanova et al., 2014). Sea ice thickness uncertainty $\sigma_H$ is provided in the AWI CS2 ice thickness product (Ricker et al., 2014). Ice drift uncertainty $\sigma_D$ is





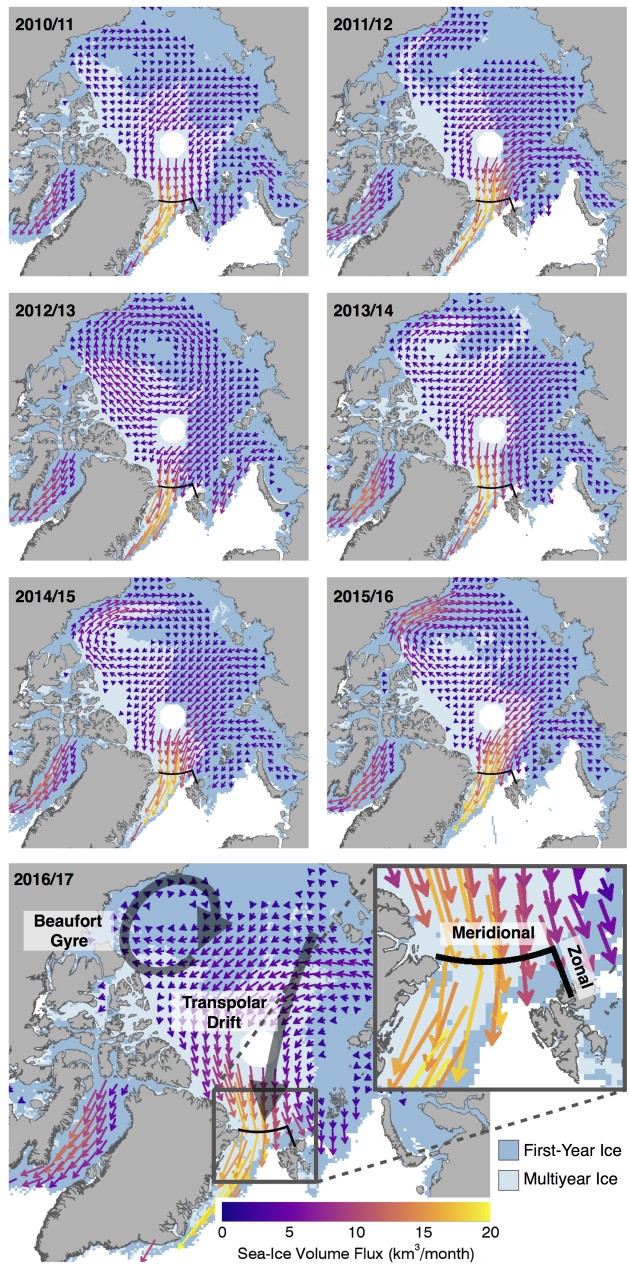

**Figure 1.** Means of Arctic sea-ice volume fluxes for 2010/2011 - 2016/2017 between October and April. The enlarged box shows the location of the Fram Strait gate at 82°N, which is used for the calculation of the export rates, separated into meridional and zonal gates.

estimated using the empirical error functions for monthly mean Arctic sea-ice drift given in Sumata et al. (2015). This study utilizes drift estimates from high-resolution SAR data as a reference. Drift uncertainties of low resolution monthly mean prod-



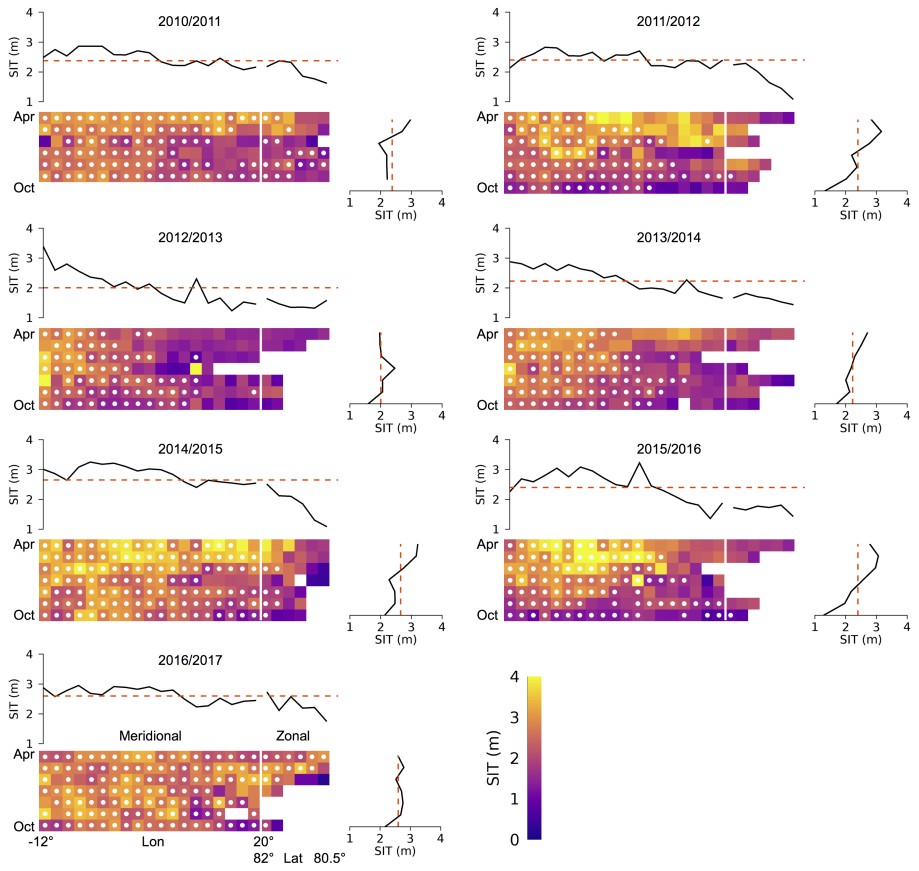

**Figure 2.** Spatiotemporal variability of sea ice thickness (SIT) at the Fram Strait gate from October to April between 2010 and 2017. Upper sub-panels show the temporal averaged SIT. Right sub-panels show the average over the gate SIT for each month within the October-April period. The white dots represent grid cells that contain multiyear ice.

ucts in x and y directions are provided for different drift speeds and ice concentrations. These uncertainties are then combined with the uncertainty of the reference ice drift. The deduced drift uncertainties for the low resolution drift products are in the range of 1.0 km/d, which is comparable to uncertainties estimated in previous studies (Spreen et al., 2009).

We obtain the total ice volume flux through the Fram Strait ($Q_{\mathrm{Ex}}$) by adding up the meridional zonal grid cell fluxes $Q_v$ and
5    $Q_u$ along the gate:

$$Q_{\mathrm{Ex}} = \sum Q_u + \sum Q_v. \tag{4}$$

Note that following the axes conventions, ice volume export $Q_{\mathrm{Ex}}$ has a negative algebraic sign, corresponding to a sea ice loss from the Arctic Basin.





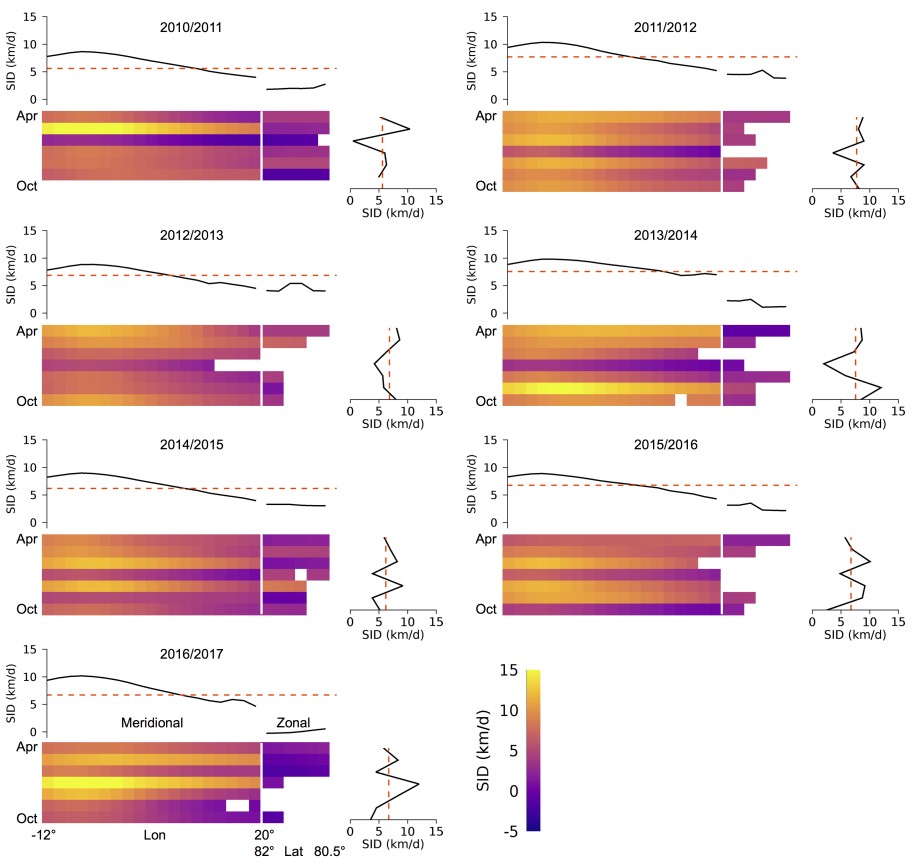

**Figure 3.** Spatiotemporal variability of OSISAF sea ice drift (SID) at the Fram Strait gate from October to April between 2010 and 2017. Upper sub-panels show the temporal average SID. Right sub-panels show the average over the gate SID for each month within the October-April period.

## 3 Results

In this section, we first examine sea ice drift, thickness and concentration at the Fram Strait gate. Throughout the study, we use the OSISAF drift as the reference product, because it shows the best performance among the used products in the Fram Strait (Sumata et al., 2014). Second, we present estimates of the ice volume flux in the Arctic and the calculated export through Fram Strait. Third, we examine the choice of the drift product, computing ice volume export using also IFREMER and NSIDC ice drift estimates. Throughout the paper, we refer to the winter period from October to April (OA). However, seasonal export estimates are calculated adding together monthly export from November to April (NA), since we have no ice thickness estimates for October 2010.





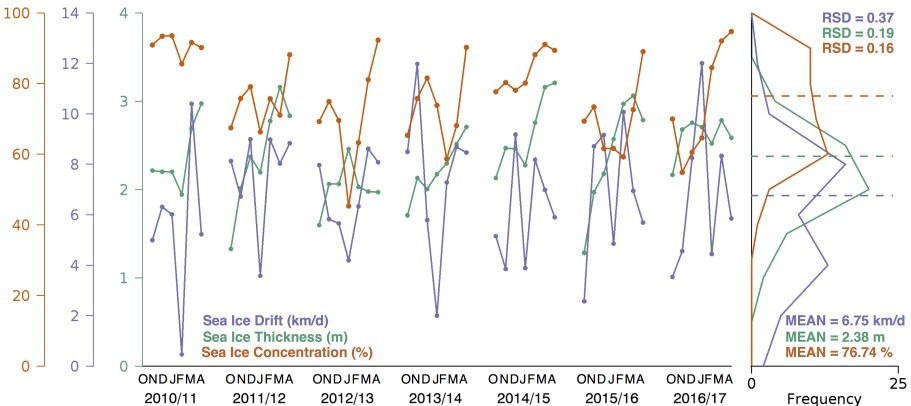

**Figure 4.** OSISAF sea ice drift, ice thickness, and ice concentration averaged over the entire Fram Strait gate, between October and April for winter seasons 2010/2011 to 2016/2017. The right box shows the corresponding histograms with the overall means and relative standard deviations (RSD).

## 3.1 Sea ice drift, thickness and concentration at the gate

We consider all input parameters for Eq. (1), sea ice thickness (H), sea ice drift (D), and ice concentration (C). Figure 2 shows the spatiotemporal distribution of CS2 ice thickness along the meridional and zonal gates through each winter season, separated into FYI and MYI. Ice thickness along the gate is variable and ranges from 0 to 5 m. The mean gate thickness reveals

a consistent gradient from thinner ice in October to thicker ice in April in all years, although the gradient can be small for some years (e.g. 2016/2017). Averaging over each OA period reveals the spatial thickness distribution along the meridional and zonal gates. In 2012/2013 and 2012/2013, we find a significant positive thickness gradient towards the coast of Greenland, while in other years, this is less pronounced. At the zonal gate, ice thickness decreases towards Svalbard. As indicated in Table 1, during winter seasons 2011/2012, 2012/2013 and 2013/2014, MYI fraction is lower compared to other years. In 2012/2013

and 2013/2014, the lack of MYI in the eastern part of the gate is replenished by FYI that is thinner than 1.5 m. In seasons 2010/2011 and 2016/2017, MYI fraction at the zonal gate is larger than in other years. In 2011/2012, from February to March, the indicated FYI is rather thick (> 2 m), similar to the indicated MYI towards the coast of Greenland.

Figure 3 shows the spatiotemporal distribution of the OSISAF ice drift along the meridional and zonal gates through each winter season. In contrast to ice thickness, the drift reveals a larger temporal variability with monthly differences of up to 10

km/d and without a distinct trend within each winter season. On the other hand, the OA period averages of the drift show a consistent spatial trend for all years, from less than 5 km/d in the east (20°E) and at the zonal gate, to a maximum of 9-10 km/d at about 6°W, followed by a decrease towards the coast of Greenland. The stationary peak at about 6°W suggests a large scale forcing and could be associated with the East Greenland Current (Rudels et al., 2002). We also notice that mean drift across the zonal gate is only 35 % of the mean drift across the meridional gate. The IFREMER and NSIDC ice drift also exhibit similar

patterns as OSISAF (not shown).



**Table 2.** Monthly Arctic sea-ice volume export through the Fram Strait in km$^3$/month, computed with OSISAF ice drift.

|         | Oct  | Nov  | Dec  | Jan  | Feb  | Mar  | Apr  |
|---------|------|------|------|------|------|------|------|
| 2010/11 | –    | -227 | -275 | -267 | -21  | -540 | -279 |
| 2011/12 | -164 | -214 | -354 | -129 | -381 | -379 | -487 |
| 2012/13 | -203 | -182 | -187 | -103 | -163 | -299 | -318 |
| 2013/14 | -215 | -400 | -231 | -78  | -195 | -345 | -452 |
| 2014/15 | -200 | -165 | -373 | -160 | -425 | -429 | -354 |
| 2015/16 | -52  | -261 | -275 | -177 | -352 | -348 | -310 |
| 2016/17 | -129 | -151 | -307 | -466 | -201 | -431 | -292 |

**Table 3.** Total Arctic sea-ice volume export through the Fram Strait for winter seasons 2010/2011 - 2016/2017, added together over the November-April period. Volume export has been computed using three different ice drift products, using $Q_{\mathrm{Ex,OSISAF}}$ as the reference product. The last column shows the fraction of exported multiyear sea ice (% MYI).

| Season  | $Q_{\mathrm{Ex}}$ OSISAF ($10^3$km$^3$) | $\Delta Q_{\mathrm{Ex}}$ IFREMER-OSISAF | $\Delta Q_{\mathrm{Ex}}$ NSIDC-OSISAF | MYI (%) |
|---------|------------------|------------------|------------------|------|
| 2010/11 | -1.61 ± 0.21     | +0.22            | +0.15            | 90   |
| 2011/12 | -1.94 ± 0.22     | +0.48            | +0.80            | 68   |
| 2012/13 | -1.25 ± 0.16     | +0.27            | +0.36            | 64   |
| 2013/14 | -1.70 ± 0.20     | +0.36            | +0.55            | 68   |
| 2014/15 | -1.91 ± 0.23     | +0.48            | +0.38            | 92   |
| 2015/16 | -1.72 ± 0.19     | +0.53            |                  | 81   |
| 2016/17 | -1.85 ± 0.21     | +0.48            |                  | 94   |

Figure 4 illustrates ice drift, thickness, and concentration averaged over the entire gate. Here, ice concentration represents the fraction of ice covered area along the entire gate, including the zonal and meridional parts. Ice concentration at the meridional gate typically ranges between 70 % and 100 %, while the zonal gate remains ice free in some areas over several months. As indicated in Figures 2 and 3, in contrast to ice drift, ice thickness shows a trend in most of the winter seasons. The same holds for

5  the ice concentration as ice extent at the zonal gate north of Svalbard increases during winter (Figures 2 and 3). In 2010/2011, the gate was almost entirely ice covered during the OA period. The histograms refer to the drift, thickness and concentration time series over the entire 7-years period. The drift distribution reveals two modes and a larger degree of dispersion than the ice thickness and concentration distribution. In order to compare and quantify the extent of variability of the three parameters we compute their relative standard deviation (RSD), which is the ratio of the standard deviation to the mean. We find that the

10  RSD of the ice drift (0.37) is roughly double of the RSD of ice thickness (0.19) and ice concentration (0.16).





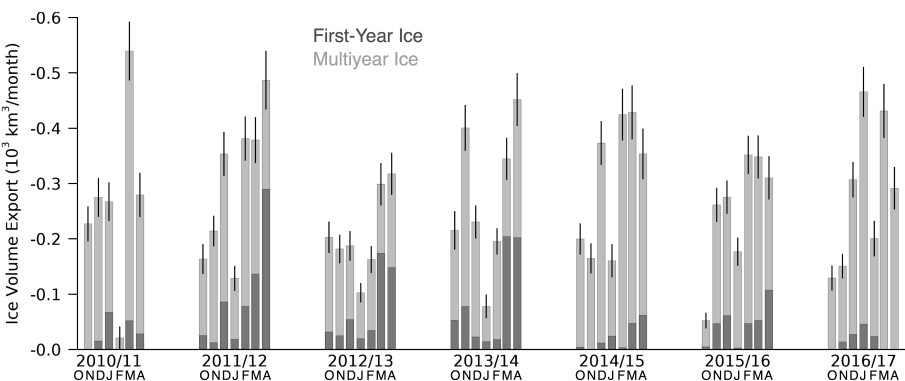

**Figure 5.** Monthly sea-ice volume export through Fram Strait from October to April for the period 2010/2011 - 2016/2017, using the OSISAF ice drift product. The volume export is devided into first- and multiyear sea ice. Uncertainties are represented by error bars. October 2010 data are missing due to unavailability of CryoSat-2 data.

## 3.2 Sea ice volume flux and export through the Fram Strait

Figure 1 shows the retrieved ice volume flux as means over the OA period for the northern hemisphere for the 7 years of the CS2 operational period (2010-2017), using OSISAF ice drift data. The two major patterns are the Beaufort Gyre and the Transpolar Drift conveying ice towards Fram Strait. There, the ice fluxes reach maximum values of 20 km$^3$/month or more, with a steep gradient along a north-south axis. MYI is mainly exported through the meridional part of the gate, while sea ice at the zonal part is primarily FYI. The monthly sea ice volume export through Fram Strait is shown in Table 2 and Figure 5. During the 7-years period, the maximum monthly ice volume export of -540 km$^3$/month occurs in March 2011, while the minimum of -21 km$^3$/month is found in February 2011. Table 3 provides the total ice volume export ($Q_{\text{Ex,OSISAF}}$) through the Fram Strait gate for the NA period. We find a maximum export of -1910 $\pm$ 230 km$^3$ for 2011/2012 and a minimum of -1250 $\pm$ 160 km$^3$ in 2012/2013. The major fraction of exported sea ice is represented by MYI. However, in few months like April 2012, the fraction of exported FYI exceeds the MYI fraction. Table 3 shows the MYI fraction averaged over the seasons. A maximum of 94 % occurs in 2016/2017 and a minimum of 64 % occurs in 2012/2013.

## 3.3 Deriving sea ice volume export using different ice drift products

Figure 6a shows an example for monthly ice volume flux through the Fram Strait gate and the contributions to the meridional and zonal parts of the gate, using three different drift products. All three retrievals exhibit consistent temporal and spatial variations along the gate, but differ in magnitude. $Q_{\text{Ex,OSISAF}}$ and $Q_{\text{Ex,IFREMER}}$ always exceed $Q_{\text{Ex,NSIDC}}$ by about 0.2-0.3 km$^3$/day. Uncertainties of the ice flux through each grid cell are in the range of 0.1 km$^3$/day. Figure 6b shows the total ice volume export through the Fram Strait during the NA period between 2010 and 2017, computed with the three different drift products. The variations of $Q_{\text{Ex,OSISAF}}$ and $Q_{\text{Ex,IFREMER}}$ correlate, but differ in magnitude. Table 3 provides the corresponding annual differences between the products. Using IFREMER ice drift, the derived ice export is 200-500 km$^3$



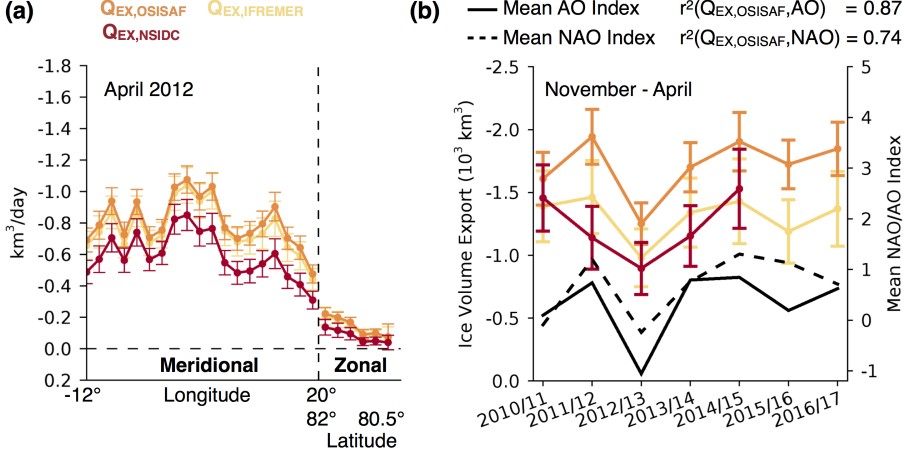

**Figure 6.** (a) Example for monthly ice volume export through the Fram Strait gate (April 2012) derived from three different ice drift products. (b) Total Arctic sea-ice volume export through the Fram Strait for winter seasons 2010/2011 - 2016/2017, added together over the November-April period and derived from three different ice drift products. Mean Arctic Oscillation (AO) Index and mean North Atlantic Oscillation Index (NAO) are shown for the same period, including coefficients of determination ($r^2$).

lower than export derived using OSISAF ice drift, which corresponds to a mean difference of about -23 %. $Q_{\text{Ex,NSIDC}}$ is the lowest among the three estimates and shows mean difference of about -26 %, relating to $Q_{\text{Ex,OSISAF}}$. Also, the interannual variability of $Q_{\text{Ex,NSIDC}}$ compared to both $Q_{\text{Ex,OSISAF}}$ and $Q_{\text{Ex,IFREMER}}$ is slightly different as $Q_{\text{Ex,NSIDC}}$ decreases from 2010/2011 to 2011/2012, while the other both retrievals show an increase. Nevertheless, the main variations with the minimum

5 in 2012/2013 and the magnitudes of spatial gradients are similar for all products. Therefore, the choice of the drift products has no major impact on our variability analysis. We also note, that it is not within the scope of this work to determine which product provides the most accurate estimate of sea ice drift in the Fram Strait.

## 4 Discussion

### 4.1 Relative contribution of sea ice drift, thickness and concentration to the volume flux variability

10 In order to understand the mechanisms behind the variability in ice volume export, we now examine the three input parameters, ice drift, thickness and concentration, in more detail. As shown in section 3.1, the thickness averaged across Fram Strait exhibits significant interannual changes with an overall increase in spring. This increase at the gate from autumn to spring can be associated with the thermodynamic ice growth and deformation of FYI and thin second-year ice. For example, In 2011/2012 and 2013/2014, thickness of FYI grid cells rises from October to April (Figure 2). In contrast, in 2016/2017, the fraction of

15 FYI passing the gate is only 6 %, and consequently, we do not observe significant changes in mean ice thickness during the OA period. Similarly, we observe an increase in mean ice concentration at the gate during the OA period. Considering the ice



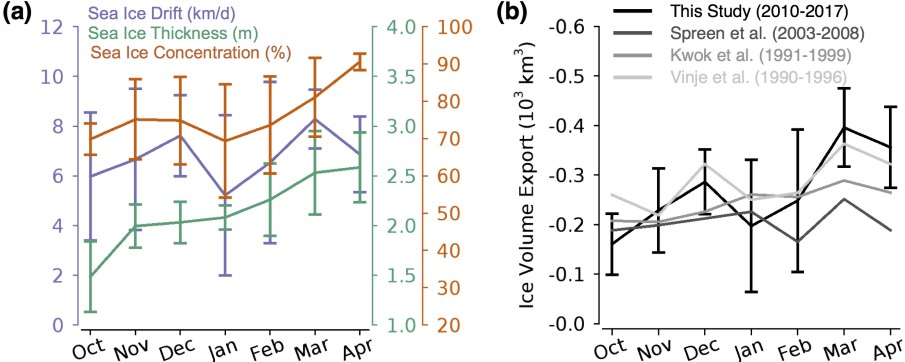

**Figure 7.** (a) Mean monthly sea ice drift, thickness and concentration at the Fram Strait gate over the years 2010/2011 - 2016/2017 with corresponding standard deviations. (b) Mean monthly ice volume export through the Fram Strait from this study, and from Spreen et al. (2009), Kwok et al. (2004), and Vinje et al. (1998) covering different periods. (b)

drift, we find opposite features. The mean monthly drift in the time domain is highly variable, without a distinct trend over the OA period (Figure 7a). These characteristics of the variability of input parameters affect the mean monthly ice volume export (Figure 7b). The mean seasonal cycle over the period 2010-2017 is characterized by minimums in October and January and the maximum in March. Considering seasonal cycles of drift, thickness, and concentration at the gate and comparing it

with the seasonal cycle of the ice export, we find that the variability is mostly explained by the ice drift (Figure 7a) as also suggested by the RSD. On the other hand, the positive gradient of the ice volume export between autumn and spring with the annual maximum in March can be associated with the seasonal cycle of sea ice thickness (Figure 7a). This seems primarily driven by thermodynamic ice growth and deformation. Although the seasonal cycle of mean ice concentration along the entire gate shows positive gradients as well, with a similar RSD as the ice thickness, it seems to play a minor role for the ice export

variability. This is because ice concentration variability at the meridional gate is small due to the persistent ice coverage over the season. Considered separately, we find a RSD of 0.78 at the zonal gate and a RSD of 0.08 at the meridional gate. But due to the smaller size, lower ice drift and thinner ice, the zonal volume flux is only about 4 % of the total ice export over the 7-years period.

### 4.2 Comparison to previous studies

Sea-ice export through the Fram Strait and its variability has been the focus of several previous studies. A major difference in the method is the choice of the position of the gate. Smedsrud et al. (2017) placed the gate at 79°N, Spreen et al. (2009) placed their most northern gate at 80°N, and Kwok and Rothrock (1999) placed their gate at about 81°N. Except for the study of Krumpen et al. (2016), all these previous studies use only a meridional gate or a straight connection between Greenland and Spitsbergen. The major advantage of using a gate positioned further north like at 82°N is that ice motion products and

thickness estimates from satellites show lower uncertainties at this latitude. Indeed, errors and biases of low resolution ice drift data derived from passive microwave and scatterometer data become larger as ice velocity increases, and velocity tends to be



larger with steeper gradients south of 82°N (Sumata et al., 2014, 2015). On the other hand, uncertainty of CS2 ice thickness increases at lower latitudes, especially in Fram Strait due to sparse orbit coverage (Ricker et al., 2014, 2017b). Therefore, we followed the approach of Krumpen et al. (2016), placing the gate at 82°N, which appears to be a good compromise in order to reduce uncertainty associated with our ice volume export estimate.

Figure 7b shows the mean monthly winter export from October to April from this study, compared to previous estimates (Kwok et al., 2004; Spreen et al., 2009; Vinje et al., 1998). Vinje et al. (1998) and Kwok et al. (2004) use upward-looking sonar (ULS) data for the estimation of ice thickness in the Fram Strait. Vinje et al. (1998) use ULS ice draft measurements from 1990-1996 in combination with buoy and SAR-based ice drift estimates, and estimate ice volume fluxes through the Fram Strait that show maxima of up to about -600 km$^3$/month. Kwok et al. (2004) investigated nearly the same period (1991-1999)
and find a maximum monthly export of -509 km$^3$/month in December 1994. Their estimates are generally lower than in Vinje et al. (1998). Spreen et al. (2009) use ice thickness and drift estimates derived from satellite data to compute ice volume flux, and therefore, their study is methodically similar to our work. However, besides the different gate at 80°N, they use a different ice thickness retrieval (ICESat) and another low resolution ice drift data set (Cersat/Ifremer, AMSR-E) to derive ice volume export. For the period 2003-2008, they estimate monthly winter ice volume export ranging from -100 to -420 km$^3$/month, using
ULS ice thickness estimates to complement the two month-long ICESat measurement periods per year. The largest difference of about 150 km$^3$ between the lowest and largest estimate is found in March. Our estimate seems to be the one with the highest change between October and April, e.g. our estimates are the lowest in October, and the highest in April (Figure 7b). Our seasonal cycle also reveals higher variability. Several factors might cause this discrepancy:

1. The observing periods are not overlapping and therefore, differences in mean monthly export can be caused by natural
variations in ice thickness and drift.

2. Bottom melt due to the recirculation of warm Atlantic water between between 82°N and 80°N might lead to a reduction in ice volume (Wekerle et al., 2017).

3. The low resolution of the drift data might lead to systematic uncertainties in the volume flux at the gate, especially near the coast and the ice margins, affecting all retrievals.

4. Systematic differences between the CS2 and ICESat ice thickness retrievals may appear because of different retrieval algorithms and different sensor characteristics.

5. Ice drift at 80°N might be underestimated due large ice velocities, which are not well captured in radiometer- and scatterometer-based drift products.

Despite these differences, estimates from different studies exhibit consistent features, such as the maximum in March. In the
following, we will discuss the interannual variability and the role of atmospheric circulation patterns.



### 4.3  Interannual ice volume export variability

The time series of winter ice volume export through the Fram Strait reveals a significant decrease of 500 km$^3$ from 2011/2012
to 2012/2013 (Figure 6). This decrease is characterized by a drop of both the mean ice drift and the thickness through Fram
Strait. Comparing the pan Arctic ice condition in both winters, a decrease of ice thickness north of Fram Strait has been reported
for 2012/2013 by Ricker et al. (2017a), and was found to be mainly a result of anomalous summer melt and late freeze up in
2012. The ice in the area north of Fram Strait is the main source of exported ice (Smedsrud et al., 2017) and thus, we also find
a drop in ice thickness at the Fram Strait for 2012/2013, which is accompanied by a lower mean drift (Figure 4). In contrast,
in winter season 2013/2014, which followed a cold Arctic summer with low melt rates (Tilling et al., 2015), ice thickness
at the gate is increasing, accompanied by a higher mean drift (Figure 4). This results in an ice volume export comparable to
2010/2011 (Figure 6).

We also examine the link between ice volume export and North Atlantic Oscillation (NAO) index and the Arctic Oscillation
(AO) index (Figure 6b). The NAO index is defined as the sea level pressure anomaly between Lisbon, Portugal, and Reykjavik,
Iceland. A positive NAO index is associated with an Icelandic low and a corresponding high-pressure system over the Azores.
When the Icelandic low is intensified, the sea level pressure gradient in the Fram Strait increases, leading to strong northerly
winds and hence, increased sea ice drift (Kwok and Rothrock, 1999; Ionita et al., 2016; Smedsrud et al., 2017). Thus, a high,
positive NAO index is associated with high ice volume export rates, since ice drift primarily drives the ice volume export
variability. If both pressure systems are weak or even reversed, the NAO phase becomes negative and correlation between
NAO and sea level pressure gradient along the Fram Strait decreases. The variability of the NAO is largest during the winter
season. We have obtained monthly NAO indices from the National Oceanic and Atmospheric Administration (NOAA) and
averaged them over the NA period. Positive phases (>1) of the NAO index occurred in 2011/2012 and 2014/2015, coinciding
with increased mean monthly ice volume export rates (Figure 6b).

The sea level pressure gradient variability through the Fram Strait is also captured in the AO and its corresponding index,
described in Thompson and Wallace (1998). The AO pattern involves an oscillation of the sea level pressure between the Arctic
basin and the surrounding zonal belt. The AO therefore includes characteristics of the NAO, which is regionally bounded. We
have obtained monthly AO indices from NOAA (http://www.cpc.ncep.noaa.gov) and averaged them over the NA period. The
variability of the AO index is similar to the variability of the NAO index and the ice volume export, especially if the flux is
computed with IFREMER and OSISAF ice drift (Figure 6b). The correlation between ice export and AO index (r$^2$ = 0.87) is
larger than with the NAO index (r$^2$ = 0.74). This is because the NAO index decreases is 2016/2017, while the ice volume export
increases compared to the previous year. However, we acknowledge that a longer time series is required to obtain statistical
meaningful correlation coefficients.

### 4.4  The impact of ice volume export on Arctic ice mass balance

Kwok et al. (1999) investigated the area balance of the Arctic Ocean perennial ice zone between October 1996 and April 1997.
Using RADARSAT data, they reported that MYI area loss can be explained almost entirely by ice export. Moreover, their





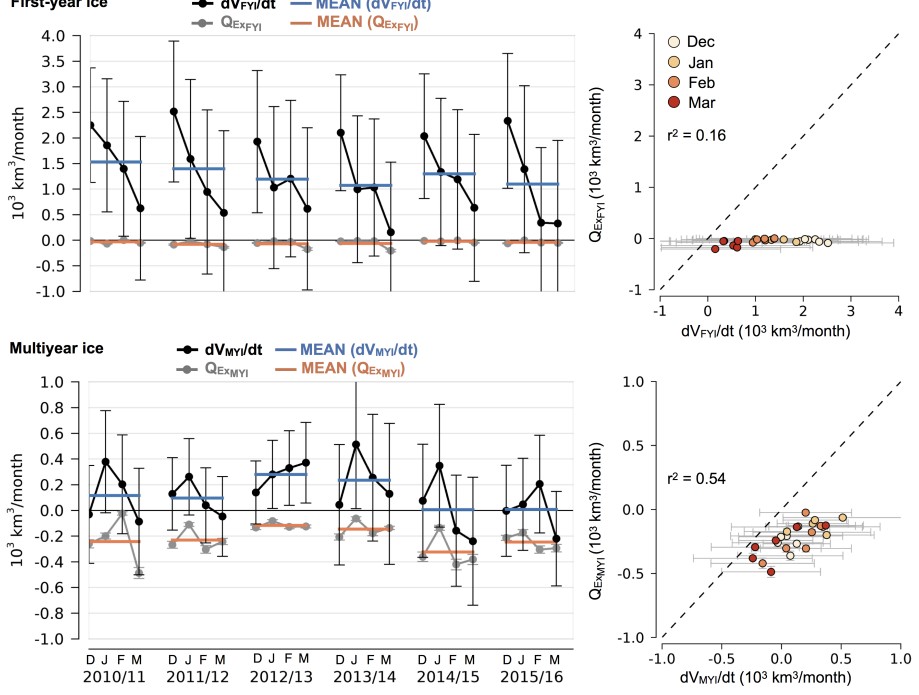

**Figure 8.** Monthly sea-ice volume export through the Fram Strait ($Q_{Ex}$) and simultaneous Arctic sea-ice volume growth rates ($dV/dt$) between December and March for the winter seasons 2010/2011 - 2015/2016, divided into first-year ice (FYI) and multiyear ice (MYI). Error bars represent the corresponding uncertainties and blue and red lines represent the means over the December-March period. Scattergrams show the relation between $dV/dt$ and $Q_{Ex}$, and corresponding coefficient of determination ($r^2$).

findings suggest that export is dominated by ice flux through the Fram Strait, while export through other gates like Nares Strait plays a minor role. According to Kwok et al. (1999), MYI area export through Nares Strait is about 5 % of the Fram Strait MYI area export. The balance of MYI area is only affected by export and ice dynamics, assuming that the net melt of MYI area is zero in winter. As a consequence, Arctic MYI area is decreasing from October to April, and in turn, this decrease is almost
5 entirely balanced by exported MYI area (Kwok et al., 1999).

In the following, we investigate the ice volume balance of the Arctic MYI. In contrast to MYI area, we assume that MYI volume growth rate of the Arctic Ocean domain ($dV_{\mathrm{MYI}}/dt$) is affected by both export ($Q_{Ex_{\mathrm{MYI}}}$) and ice volume gain due to thermodynamic growth ($dV_{\mathrm{therm_{MYI}}}/dt$). Neglecting net melt of MYI in winter, we can write:

$$\frac{dV_{\mathrm{MYI}}}{dt} = Q_{Ex_{\mathrm{MYI}}} + \left( \frac{dV_{\mathrm{therm_{MYI}}}}{dt} + \frac{dV_{\mathrm{resid_{MYI}}}}{dt} \right). \tag{5}$$

10 The term $dV_{\mathrm{resid_{MYI}}}/dt$ accounts for residual contributions, such as ice deformation that might change the bulk ice density, which we assume to be constant. However, we believe that $dV_{\mathrm{resid_{MYI}}}/dt$ is small compared to $dV_{\mathrm{therm_{MYI}}}/dt$. Therefore, we assume that winter MYI volume variability is primarily affected by changes in ice volume export and thermodynamic growth.



We estimate monthly Arctic FYI volume growth $dV_{\mathrm{FYI}}/dt$ and monthly Arctic MYI volume growth ($dV_{\mathrm{MYI}}/dt$) by a 3-point Lagrangian interpolation scheme, where we exclude all ice south of the Fram Strait gate. Since CS2 ice thickness data are not available in October 2010, we compute $dV_{\mathrm{FYI}}/dt$ and $dV_{\mathrm{MYI}}/dt$ using data over the NA period and therefore obtain $dV_{\mathrm{FYI}}/dt$ and $dV_{\mathrm{MYI}}/dt$ values for December to March. Figure 8 shows monthly $dV_{\mathrm{FYI}}/dt$ ($dV_{\mathrm{MYI}}/dt$) and corresponding

$Q_{\mathrm{Ex_{FYI}}}$ ($Q_{\mathrm{Ex_{MYI}}}$) for 6 years. Export has no significant impact on FYI volume variability, because the variability is dominated by the gain due to thermodynamic ice growth, which is up to 15 times higher than $Q_{\mathrm{Ex_{FYI}}}$. The Scattergram in Figure 8 reveals the seasonal cycle of $dV_{\mathrm{FYI}}/dt$, with a decrease of ice growth from December to March.

MYI volume growth $dV_{\mathrm{MYI}}/dt$ does not follow a seasonal cycle as it is the case for FYI volume growth $dV_{\mathrm{FYI}}/dt$ that is primarily driven by the thermodynamic ice growth (Ricker et al., 2017a). Our estimated $dV_{\mathrm{FYI}}/dt$ is in the order of 10 times

higher than MYI volume growth $dV_{\mathrm{MYI}}/dt$. This is a result of the thicker MYI associated with reduced thermodynamic ice growth on the one hand, and new forming ice as well as faster growing FYI on the other hand. It appears that $Q_{\mathrm{Ex_{MYI}}}$ is just in the range to almost or entirely balance the volume gain of the second term in Eq. (5), ($dV_{\mathrm{therm_{MYI}}}/dt + dV_{\mathrm{resid_{MYI}}}/dt$). For example, in 2014/2015 and 2015/2016, mean $dV_{\mathrm{MYI}}/dt$ is nearly zero due to a large $Q_{\mathrm{Ex_{MYI}}}$ between December and March. As a consequence, the variability of $dV_{\mathrm{MYI}}/dt$ is significantly driven by $Q_{\mathrm{Ex_{MYI}}}$, revealing a coefficient of determination ($r^2$)

of 0.54, which means that 54 % of $dV_{\mathrm{MYI}}/dt$ during winter can be explained by variations of $Q_{\mathrm{Ex_{MYI}}}$. From that, we can deduce that the variability of $dV_{\mathrm{MYI}}/dt$ is significantly driven by the variability of the ice drift in the Fram Strait.

The high correlation (0.74) between $Q_{\mathrm{Ex_{MYI}}}$ and $dV_{\mathrm{MYI}}/dt$ is also noticeable. This proves the accuracy of Arctic MYI volume estimates as the correlation between $Q_{\mathrm{Ex_{MYI}}}$ and $dV_{\mathrm{MYI}}/dt$ exposes the signal of ice volume export in the MYI volume budget. In case of large errors in $dV_{\mathrm{MYI}}/dt$ as indicated in Figure 8 by the error bars, correlation with $Q_{\mathrm{Ex_{MYI}}}$ would

be degraded.

## 5   Conclusions

Here we have used, for the first time, the CryoSat-2 ice thickness retrievals in order to quantify the sea ice export through Fram Strait. We performed a detailed analysis of variability and important processes for the Arctic multiyear ice (MYI) mass balance. Based on our analysis, the following conclusions can be drawn:

1. Based on different ice drift products, the three ice volume export retrievals ($Q_{\mathrm{Ex,OSISAF}}$, $Q_{\mathrm{Ex,IFREMER}}$, $Q_{\mathrm{Ex,NSIDC}}$) exhibit similarities in their variability, although they differ in magnitude by -23 % ($Q_{\mathrm{Ex,IFREMER}}$) and -26 % ($Q_{\mathrm{Ex,NSIDC}}$), compared to $Q_{\mathrm{Ex,OSISAF}}$. In order to investigate long-term trends in ice volume export derived from multiple satellite observations, we therefore need to construct multi-sensor consistent time series of ice drift, thickness, and concentration. Moreover, a consistent methodology to compute ice volume flux through Fram Strait is required.

2. Ice drift shows coherent spatial variability across Fram Strait, but high frequency variability from month to month. The mean monthly ice drift across Fram Strait shows a peak at about 6°W, which could be associated with the East Greenland Current.



3. The relative standard deviation (RSD) is a measure to compare the variability of different physical quantities. At the Fram Strait gate, RSD of ice drift (0.37) is roughly twice as high as the RSD of ice thickness (0.19) and concentration (0.16) for the observation period of 2010/2011-2016/1017, revealing that ice drift is the main driver of seasonal and interannual variability of ice volume export. However, the seasonal trend of ice volume export is driven by variations in ice thickness due to the thermodynamic growth that typically leads to a maximum in March. Ice concentration variability is large at the zonal gate (RSD = 0.78), but small at the meridional gate (RSD = 0.08), where 96 % of the sea ice is exported.

4. The interannual variations of ice volume export can be explained by large scale variability of the atmospheric circulation captured by the Arctic Oscillation and North Atlantic Oscillation indices.

5. While the seasonal cycle of Arctic first-year ice volume is driven by thermodynamic ice growth, 54 % of the variability of Arctic MYI volume over the December-March period can be explained by ice volume export through the Fram Strait.

6. While MYI area declines during the seasonal cycle, MYI volume is in equilibrium or slightly increases. We believe that this is a consequence of thermodynamic ice growth, which compensates the loss due to ice export. Contrary, MYI area loss due to export adds to the loss by area compression due to convergence.

*Data availability.* Sea ice concentration, sea ice type and sea ice drift data are provided by OSISAF (http://osisaf.met.no). CryoSat-2 ice thickness data from 2010-2017 are provided by http://www.meereisportal.de. IFREMER ice drift data from 2010 to 2017 are provided via CERSAT (http://cersat.ifremer.fr). NSIDC ice drift data from 2010 to 2015 are provided via https://nsidc.org.

*Author contributions.* Robert Ricker conducted the ice volume flux calculations and the analysis. Fanny Girard-Ardhuin, Thomas Krumpen and Camille Lique contributed to the analysis of the ice volume flux data. Robert Ricker wrote the paper and all Co-authors contributed to the discussion and gave input for writing.

*Competing interests.* The authors declare no conflict of interest.

*Acknowledgements.* This work has been conducted in the framework of the project: Space-borne observations for detecting and forecasting sea ice cover extremes (SPICES) funded by the European Union (H2020) (Grant: 640161). CryoSat-2 data are provided by http://www.meereisportal.de (Grant REKLIM-2013-04). Moreover, this work was supported by the German Ministry for Science and Education under Grant 03F0777A (QUARCCS) and the CORESAT project funded by the Norwegian Research Council (grant 222681).





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
