# Peer review of "Satellite-derived sea ice export and its impact on Arctic ice mass balance"

_The Cryosphere, 2018_

## Referee Comment (RC1) · Anonymous Referee #1 · 26 Mar 2018

This study combines sea ice thickness retrievals from CryoSat-2 with different ice drift products to estimate the volume of ice export through the Fram Strait over the winters of 2010-2017. The authors find that ice drift variability dominates the variability of ice volume export over annual and inter-annual timescales, but the seasonal cycle is also impacted by the thickness of exported ice. The export of sea ice through the Fram Strait accounts for 54% of the variability of multiyear ice (MYI) volume over a given winter season.

The manuscript is clearly written and the figures are well-constructed and informative. Unfortunately, I still struggled with this review as I'm left wondering what the key purpose of the paper is. The manuscript includes a wealth of information but doesn't read like a complete method or scientific study. This is highlighted by the concluding bullets

ranging from comparison of drift products to importance of ice export, presented as a list rather than a logical connected paragraph.

For a methods-based paper I would expect a more thorough description of the product development. This includes expansion on the error analysis explaining why the specific approach was chosen, how sea ice drift uncertainty is estimated using empirical error functions (brief summary of Sumata (2015) method), which high resolution SAR data is used a reference, and why such a reference is needed. The usefulness of the paper in a scientific sense is currently limited over such a short time frame, and it lacks novelty considering the number of existing sea ice export studies for the Fram Strait. The obvious way to develop the paper scientifically would be to investigate long-term trends in ice volume export, but as the authors state this would require a consistent methodology to compute ice volume flux through Fram Strait from multiple products.

I encourage the authors to think about their intended purpose for the paper then either a.) sufficiently describe the development of their new Arctic sea ice volume export product or b.) expand their scientific analysis utilising the product. Despite these reservations I would like to repeat that this was a well-written paper and the content will be of interest to the sea ice community, so I have included some detailed and technical comments below.

Detailed comments

P1L18-P2L7: The reasoning is not clear here with regards to concentrating on MYI and winter. For example, the authors should explicitly state that winter does not play such an important role for FYI mass balance, and why. They also mention summer ice concentration when the focus of the manuscript is on the winter period. If the authors want to justify their concentration on a given ice type and season then I suggest they first discuss winter ice mass balance variations (MYI and FYI) and then summer (MYI and FYI), then reach a logical conclusion.

P2L16: State ICESat periods

P2L20: "...we use the CS-2 ice thickness dataset..." -> "...we use \*our\* CS-2 ice thickness dataset..." There are numerous datasets, so the authors should be specific about which is used.

P2L21: Be more explicit about which part of the study is novel (i.e. the "first" estimates of what). It is not the sea ice export estimates themselves, but the timeframe for which they're provided.

P2L21: Define "winter"

P4L21-23: Explain how unconstrained polynomials are dealt with at lower latitudes

P4L26: And also snow depth, correct?

P5L11-12: NSIDC products are also provided monthly

P6 Figure 1: The FYI and MYI masks are quite hard to distinguish with the current colour separation. I'd like to be able to see them clearly for each year.

P9 Figure 4: It is not clear why the frequency scale ranges from 0-25 for the right hand box. It's also hard to see the variation in the lines over one another. Sub-plots could work better here. It may be for thickness and drift the time-series isn't necessary, as the relevant data is already displayed in Figures 2 and 3.

P9L7: "2012/2013" is repeated. I believe second date should be 2014/2015.

P12L5-6: I suggest moving the statement that the choice of drift product has no major impact on the variability analysis to the start of the start of the section, as until then I wasn't sure of the point of the section.

P14L1: Should this read "Similarly" rather than "On the other hand" as it's previously explained that uncertainty of ice drift also increases at lower latitudes.

P14L3: What is the "compromise" here? Uncertainty reduction vs. discarding higher ice velocities? It's not clear.

P15L33: "...**seasonal** or **winter** MYI area loss can be explained almost entirely by ice export."

Technical comments

P1 L1: "*Sea* ice volume export..."

P2L18: "sea-ice" -> "sea ice" for consistency

P2L22: "access" -> "assess"

P3L11: "...(Continuous MCC)..." -> "...(Continuous Maximum cross-correlation (MCC))..."

P3L9: "Table 1" -> "Table 3"

---

## Referee Comment (RC2) · Anonymous Referee #2 · 16 Apr 2018

The manuscript presents a new time series of Fram Strait sea ice volume export for the years 2010 to 2017. Fram Strait is the main gateway for sea ice leaving the Arctic and therefore estimates at that gate are a good indicator for sea ice mass change by ice export. The topic therefore is highly relevant for Arctic climate understanding. The authors describe a method solely based on satellite data, i.e., CryoSat-2 and microwave radiometer and scatterometer data. A similar method was applied before for different satellites but not for this combination and more recent years. While not discussed here the method potentially can serve as a tool to extend previous ice volume export time series. Results are discussed in connection with atmospheric forcing (NAO, AO) and the total Arctic mass balance. The topic is suitable for publication in The Cryosphere.

I, however, have some mayor concerns, which have to be addressed beforehand.

[Figure]

Mayor criticism:

- Flux calculations (eq. 2 & 3) seem to contain an error (varying length of grid cell not taken into account), which can cause the volume flux to be biased low by up to 40%. This has to be corrected or justified why the flux calculation is correct as it is given. This error will change the magnitude but not the variability of all calculation. Thus most conclusions will still be valid.

- The Sea ice export estimates based on three different ice drift datasets do not agree within their uncertainty estimates. Which means that either the uncertainty estimates are wrong (to conservative) or some justification should be provided which dataset is more trustworthy. Otherwise the reader cannot use the information provided in a meaningful why. Unfortunately the difference is not just a bias but in some years exhibits different variability (Fig. 6b).

- Explanation of changes of MYI volume in the Arctic basin (4.4) does not sound physical to me. 4 out of 6 years show a gain of MYI ice volume through winter (100-300km3/month), even after taking the ice export into account. The authors attribute that to thermodynamical growth. This would mean that in most cases ice growth for MY dominates the MYI ice volume change over ice export. I find that highly unlikely. The thermodynamic growth of snow covered MYI ice of >1.5m should be close to zero. Ice export through Fram Strait should by far dominate the month to month changes. The authors need to analysis this in more detail or provide more evidence. Actually, I assume their finding are dominated by the uncertainty of their MYI classification. They only use a binary MYI/FYI mask. The increase of MYI ice volume they observe could be well not MYI but FYI that growth in the leads or is otherwise integrated within the MYI within a 25 km grid cell. In summary, I don't think their conclusion that MYI volume is increasing in most months during winter is correct. Sea ice export should dominate the MYI volume change and cause it to be negative almost always.

Detailed comments: p1,l23: better split in two sentences. Sounds like the definition of

MYI is connected to ice export.

p2,l1: add a sentence explaining how storms reduce sea ice. p2,l4: "Multiple" you only name one. p2,l13: what do you mean by "parametrization"? These studies were based on ULS ice thickness measurements.

p3, Table 1: the table should also include the name used for the three products in the text, i.e., OSISAF, IFREMER, NSIDC

p4/5, 2.3: Please discuss potential errors of MYI classification. You make quite strong use of the MYI dataset throughout this paper. However, in the convergent zone of Fram Strait, where ice gets deformed and broken up in smaller floes, ice type identification gets less reliable (surface scattering can dominate the volume scattering used for MYI type identification). Uncertainty estimates should be mentioned here and also more critically discussed later in the paper (e.g. in 3.1) when the ice types are analysed.

p5, l13: if you average them you wouldn't get the monthly displacement but the mean 48h displacement. p5, l19: hm, the gates are not aligned with the grid. The gates then would be not smooth lines like in the figure but step-wise functions, right? I think that makes flux calculations unnecessary complicated (see below). p5, eq. 2: why is l kept constant at 25km? Depending on the direction of the meridional or zonal component l can increase to sqrt(2)*l = 1.4*l=35km at 45° or not? p5, eq. 3: again, l should not be constant. With 40% the changes in effective length of your grid cell, which you did not take into account, could well dominate your error. p6,l1 and ff: Do you apply the error functions from that paper to your datasets or do you use their values? Do these error values then vary in space and time? Shouldn't the error function depend on the drift dataset or dis Sumata evaluate all your three drift datasets? Please provide some more information and clearer description of method used here. p6, Figure 1: make figure larger (full page), arrows are hard to see.

p7, Figure 2: make larger

p8, Figure 3: make larger

p9, l8/9: Do you mean Table 3? p9, l18: this is definitely related with the EGC location, which floes along the shelf edge (haven't checked but which probably is at 6W at that latitude then). See e.g. papers by de Steur et al.

p10, l5: SIC is not shown in Figs 2 & 3. What do you mean? p10, Table 3: Explain what % MYI means (maybe better in the text). Is a grid cell MYI if there was any MYI detected within the month or does it have to be >50%? What are the uncertainties of this MYI % values? Also, how does the MYI product define the MYI ice type? Is >50% MYI fraction within a grid cell considered MYI or 100% etc.?

p11, l4: here you are reporting on per grid cell values. That should be mentioned. Because without the grid size of 25km these values are quite hard to set into relation for the common reader. p11, l10: see comment for Table 3. Explain better what "majority fraction MYI" means.

p11, l19. What are the correlation coefficients? Actually, the exports correspond less from what I would have expected from Fig. 6a. Please explain in more detail why, e.g., NSIDC in 11/12 goes down while the other two go up.

p12, l6: I am not sure I agree with that conclusion. If NSIDC goes down in 11/12 while the other two go up, the variability is quite different, or? Similar for 13/14 to 14/15, where OSISAF & NSIDC show strong increase and IFREMER is more neutral. p12, l7: It is understood that you do not make a ice drift dataset validation study. However, your export estimates do not agree within their uncertainty estimates. Which means either your uncertainty estimates are wrong or you need to justify why you trust one dataset more than another. To me also the inter-annual variability in Fig. 6b is quite different for the three products. Some explanation for that should be added. p12, l13: in

p13, Figure 7: shades of gray are hard to discern; caption: remove (b) at the end.

p14, l1: Not "on the other hand". This argument also supports moving the gate north.

p14, l19/20: this is supported by the overlap of STD of this study with previous studies for most months. p14, l22/23: are there estimates of ice thickness gradient between 80 and 82°N? What gradient does CS2 show? Can you estimate the thickness gradient to support this argument? p15, l15: Have also a look at Kwok et al. (2013), which analyses ice area export in connection to different atmospheric indices (AO, DA).

p17, l5: what happened to the 7th year 16/17? p17, l6: "Scattergram" p17 l10: See my mayor comment at the top. I don't think this is correct. dV_MYI/dt should almost never be positive. p17, l15. hm, that is maybe correct. The word "variations" is not very well defined, maybe better standard deviation? However, only 29% of the variance of dV_MYI/dt is explained by Q_MYI. p17, 26: yes, there are some similarities in their variability but actually their variability differs quite significantly and they do not agree within their uncertainty estimates. I find this conclusion too positive or at least need some explanation of the problems. Having results that do not agree within their uncertainty but not to mention that I do not find acceptable. Actually, I would prefer that you guide the reader which estimate they should use or you have to increase the uncertainty estimates.

p18, l7: "explained" to what degree? Give numbers. p18, l9: How is "variability" defined if quantitative numbers are given here? p18, l11-13: I do not agree with this point. See explanations above.

p19,l17: please provide information how and where to obtain this user guide.

---

## Author Response (AR1)

**Response letter to Anonymous Referee #1**

This study combines sea ice thickness retrievals from CryoSat-2 with different ice drift products to estimate the volume of ice export through the Fram Strait over the winters of 2010-2017. The authors find that ice drift variability dominates the variability of ice volume export over annual and inter-annual timescales, but the seasonal cycle is also impacted by the thickness of exported ice. The export of sea ice through the Fram Strait accounts for 54% of the variability of multiyear ice (MYI) volume over a given winter season.

The manuscript is clearly written and the figures are well-constructed and informative. Unfortunately, I still struggled with this review as I'm left wondering what the key pur- pose of the paper is. The manuscript includes a wealth of information but doesn't read like a complete method or scientific study. This is highlighted by the concluding bullets ranging from comparison of drift products to importance of ice export, presented as a list rather than a logical connected paragraph. For a methods-based paper I would expect a more thorough description of the product development. This includes expansion on the error analysis explaining why the specific approach was chosen, how sea ice drift uncertainty is estimated using empirical error functions (brief summary of Sumata (2015) method), which high resolution SAR data is used a reference, and why such a reference is needed. The usefulness of the paper in a scientific sense is currently limited over such a short time frame, and it lacks novelty considering the number of existing sea ice export studies for the Fram Strait. The obvious way to develop the paper scientifically would be to investigate long-term trends in ice volume export, but as the authors state this would require a consistent methodology to compute ice volume flux through Fram Strait from multiple products.

I encourage the authors to think about their intended purpose for the paper then either a.) sufficiently describe the development of their new Arctic sea ice volume export product or b.) expand their scientific analysis utilizing the product. Despite these reservations I would like to repeat that this was a well-written paper and the content will be of interest to the sea ice community, so I have included some detailed and technical comments below.

We thank the reviewer for these thoughtful comments which helped to improve the paper. The novelty of the paper is to use the CryoSat-2 monthly ice thickness estimates for the computation of ice volume export. This hasn't been done before. Monthly retrievals allow a much more detailed analysis of the seasonal cycle and how the variabilities in sea-ice thickness, drift and concentration affect the volume export. These investigation were not (or very limited) possible with ICESat ice thickness retrievals that are available only twice per season (October-November and February-March), leaving gaps in December/January.

The Arctic research community is putting a lot of effort into quantifying contributions to exchanges between the Arctic and the North Atlantic (see ASOF for example, http://asof.awi.de ). One key question is: What are the fluxes of mass, heat, liquid freshwater and ice from the Arctic Ocean into the subpolar North Atlantic (ASOF II Objectives)? Our study addresses some of these issues.

We have emphasized the scientific importance of this study in the revision. Nevertheless, we also included a more detailed description of the error estimates.

We have revised the paper substantially to address the comments from both reviews. The main changes are listed in the following:

- We have carefully revised the computation of the ice volume flux and checked the results.

- We have updated the NSIDC ice drift data set, since it is now available until February 2017, which allowed us to update the NSIDC ice volume export time series until the 2015/2016 season. 2016/2017 has not been computed, since March and April are still missing in the NSIDC data.

- The Methods section has been revised for a more detailed description of the ice volume export calculations, especially of the uncertainty estimates.

- We have added a figure showing the ice concentration along the Fram Strait Gate, corresponding to Figures 2 and 3 for sea ice thickness and sea ice drift.

- We have revised section 4.4, discussing the impact of openings in the MYI zone, which might lead to a positive bias in MYI growth rates due to erroneously classified MYI.

In the following, please find our responses separately for each of the detailed comments:

Detailed comments

P1L18-P2L7: The reasoning is not clear here with regards to concentrating on MYI and winter. For example, the authors should explicitly state that winter does not play such an important role for FYI mass balance, and why. They also mention summer ice concentration when the focus of the manuscript is on the winter period. If the authors want to justify their concentration on a given ice type and season then I suggest they first discuss winter ice mass balance variations (MYI and FYI) and then summer (MYI and FYI), then reach a logical conclusion.

In this paragraph, it is our intention to point on the importance of MYI for the Arctic ice mass balance and for the state of Arctic sea ice. Winter does indeed play an important role for FYI mass balance. However, in this study, we focus on the MYI export and its effect on MYI mass balance. We have revised this paragraph for clarification.

P2L16: State ICESat periods

We added the ICESat periods (October/November and February/March)

P2L20: "...we use the CS-2 ice thickness dataset..." -> "...we use *our* CS-2 ice thickness dataset. . ." There are numerous datasets, so the authors should be specific about which is used.

Agreed. We have added „AWI" here.

P2L21: Be more explicit about which part of the study is novel (i.e. the "first" estimates of what). It is not the sea ice export estimates themselves, but the timeframe for which they're provided.

The novel part is the time frame, but also, and even more, using CS-2 ice thickness data for the first time to estimate volume export rates. The CS-2 data allow for a much more detailed analysis of seasonal variability of ice volume export as it provides monthly estimates, in contrast to ICESat, which provided only two estimates per year. We have edited the introduction to emphasis the goals and the novelty of this paper.

P2L21: Define "winter"

We have added „(October-April)" for clarification.

P4L21-23: Explain how unconstrained polynomials are dealt with at lower latitudes

Here, we consider the Warren climatology (W99) only in the Arctic Basin, e.g. where the polynomials are constraint by in situ measurements. For example, W99 shouldn't be applied over Baffin Bay, or at least with caution.

P4L26: And also snow depth, correct?

Thank you, this is indeed missing here. Added.

P5L11-12: NSIDC products are also provided monthly

Added in the text.

P6 Figure 1: The FYI and MYI masks are quite hard to distinguish with the current color separation. I'd like to be able to see them clearly for each year.

We have enlarged the figures and slightly enhanced the contrast between the blue tones, so it should be easier to distinguish now.

P9 Figure 4: It is not clear why the frequency scale ranges from 0-25 for the right hand box. It's also hard to see the variation in the lines over one another. Sub-plots could work better here. It may be for thickness and drift the time-series isn't necessary, as the relevant data is already displayed in Figures 2 and 3.

We have changed this figure. Ice thickness, concentration and drift are now divided by their means for better comparison of their variability. The frequency refers to the number of samples that are within a given bin size of 0.2.

P9L7: "2012/2013" is repeated. I believe second date should be 2014/2015.

Thank you for pointing on this. The second date should have been 2013/2014. Corrected.

P12L5-6: I suggest moving the statement that the choice of drift product has no major impact on the variability analysis to the start of the start of the section, as until then I wasn't sure of the point of the section.

The point is that this statement is a result of the comparison between the different products. Therefore, we think that this statement should be somehow at the end of this section. However, we agree that the purpose of this comparison should be mentioned in the beginning. This has been added.

P14L1: Should this read "Similarly" rather than "On the other hand" as it's previously explained that uncertainty of ice drift also increases at lower latitudes.

This is true and we have exchanged "On the other hand" with „In addition".

P14L3: What is the "compromise" here? Uncertainty reduction vs. discarding higher ice velocities? It's not clear.

There is actually no compromise, therefore we changed this sentence to avoid confusion. In the western part, towards the coast of Greenland, the shelf area is more narrow than further south. Therefore, ice velocities are quite high also close to the coast, which might be associated with uncertainties in the low resolution drift product. On the other hand, further south, e.g. 80°N, the open ocean in the eastern part of the gate is associated with uncertainties in ice concentration and velocity as well. In any case, velocities at 82°N are more reliable than further south, as shown in Sumata et al. (2014).

P15L33: ". . .**seasonal** or **winter** MYI area loss can be explained almost entirely by ice export."

„winter" added.

Technical comments

P1 L1: "*Sea* ice volume export. . ."

Fixed.

P2L18: "sea-ice" -> "sea ice" for consistency

We have corrected this and avoided hyphenating throughout the paper.

P2L22: "access" -> "assess"

Fixed.

P3L11: "...(Continuous MCC)..." -> "...(Continuous Maximum cross-correlation (MCC)). . ."

**Response letter to Anonymous Referee #2**

The manuscript presents a new time series of Fram Strait sea ice volume export for the years 2010 to 2017. Fram Strait is the main gateway for sea ice leaving the Arctic and therefore estimates at that gate are a good indicator for sea ice mass change by ice export. The topic therefore is highly relevant for Arctic climate understanding. The authors describe a method solely based on satellite data, i.e., CryoSat-2 and microwave radiometer and scatterometer data. A similar method was applied before for different satellites but not for this combination and more recent years. While not discussed here the method potentially can serve as a tool to extend previous ice volume export time series. Results are discussed in connection with atmospheric forcing (NAO, AO) and the total Arctic mass balance. The topic is suitable for publication in The Cryosphere.
I, however, have some mayor concerns, which have to be addressed beforehand.

We thank the reviewer for these thoughtful comments which helped to improve the paper. We have revised the paper substantially to address the comments from both reviews. The main changes are listed in the following:

- We have carefully revised the computation of the ice volume flux and checked the results.

- We have updated the NSIDC ice drift data set, since it is now available until February 2017, which allowed us to update the NSIDC ice volume export time series until the 2015/2016 season. 2016/2017 has not been computed, since March and April are still missing in the NSIDC data.

- The Methods section has been revised for a more detailed description of the ice volume export calculations, especially of the uncertainty estimates.

- We have added a figure showing the ice concentration along the Fram Strait Gate, corresponding to Figures 2 and 3 for sea ice thickness and sea ice drift.

- We have revised section 4.4, discussing the impact of openings in the MYI zone, which might lead to a positive bias in MYI growth rates due to erroneously classified MYI.

In the following, please find our responses separately for each of your comment:

Mayor criticism:
- Flux calculations (eq. 2 & 3) seem to contain an error (varying length of grid cell not taken into account), which can cause the volume flux to be biased low by up to 40%. This has to be corrected or justified why the flux calculation is correct as it is given. This error will change the magnitude but not the variability of all calculation. Thus most conclusions will still be valid.

We took into account the varying length of the grid cell. In Eq. 2 and 3, the grid cell length l=25 km is divided by cos(lambda), where lambda represents the longitude. Therefore, the varying grid cell length is l_uv = l/cos(lambda). We have clarified this in the text.

- The Sea ice export estimates based on three different ice drift datasets do not agree within their uncertainty estimates. Which means that either the uncertainty estimates are wrong (to conservative) or some justification should be provided which dataset is more trustworthy. Otherwise the reader cannot use the information provided in a meaningful why. Unfortunately the difference is not just a bias but in some years exhibits different variability (Fig. 6b).

We applied the uncertainty estimation according to the drift error function given in Sumata et al. (2015). For a better understanding we have included the applied equations in the revised version of this paper. However, this drift error function does not contain biases or systematic errors. These have been investigated separately in Sumata et al. (2015). We have added a paragraph to better explain the error estimates and potential biases.

- Explanation of changes of MYI volume in the Arctic basin (4.4) does not sound physical to me. 4 out of 6 years show a gain of MYI ice volume through winter (100- 300km3/month), even after taking the ice export into account. The authors attribute that to thermodynamical growth. This

would mean that in most cases ice growth for MY dominates the MYI ice volume change over ice export. I find that highly unlikely. The thermodynamic growth of snow covered MYI ice of >1.5m should be close to zero. Ice export through Fram Strait should by far dominate the month to month changes. The authors need to analysis this in more detail or provide more evidence. Actually, I assume their finding are dominated by the uncertainty of their MYI classification. They only use a binary MYI/FYI mask. The increase of MYI ice volume they observe could be well not MYI but FYI that growth in the leads or is otherwise integrated within the MYI within a 25 km grid cell. In summary, I don't think their conclusion that MYI volume is increasing in most months during winter is correct. Sea ice export should dominate the MYI volume change and cause it to be negative almost always.

We agree that section 4.4 was lacking a discussion of potential errors due to openings and forming of new ice within the MYI zone that are not well captured by the ice type product. This indeed contributes to the residual term in Eq. 5. Therefore, we have added a paragraph for clarification. However, we are convinced that thermodynamic ice growth still plays a role for MYI. It is true that 2 m thick snow covered sea ice does not show relevant thermodynamic growth anymore, but thickness of second-year or third-year sea ice can decrease to 1 m during summer melt. Then, during the freeze-up, ice grows again, even if slowly, until it reached ~2 m. Buoy measurements from the Arctic Basin do capture this behavior (Figure R1). This buoy data set covers two freezing seasons from August 2013 to August 2015 and shows how thickness of second/third year sea ice decreased during summer melt and increases again during winter. To conclude, both effects (bias due to openings in the MYI zone + thermodynamic ice growth) most likely play a role and are therefore represented in Eq.5. It is difficult to separate the two effects since quantification is rather difficult and not within the scope of this study.

[Figure]

Figure R1: Snow/ice and bottom surface for ice mass balance buoy 2013F obtained from http://imb-crrel-dartmouth.org/imb.crrel/2013F.htm.

Detailed comments:

p1,l23: better split in two sentences. Sounds like the definition of MYI is connected to ice export.
Done. We also have restructured this part of the introduction.

p2,l1: add a sentence explaining how storms reduce sea ice.

We have added a sentence on this topic referencing a study by Parkinson et al., in which they have investigated the sea ice minimum in 2012.

p2,l4: "Multiple" you only name one.

Thank you for pointing that out. Corrected. We only refer to one study here.

p2,l13: what do you mean by "parametrization"? These studies were based on ULS ice thickness measurements.

„Parametrization" here means that thickness across Fram Strait is estimated using ULS ice thickness measurements from distinct locations in the Fram Strait.

p3, Table 1: the table should also include the name used for the three products in the text, i.e., OSISAF, IFREMER, NSIDC

Yes. Fixed.

p4/5, 2.3: Please discuss potential errors of MYI classification. You make quite strong use of the MYI dataset throughout this paper. However, in the convergent zone of Fram Strait, where ice gets deformed and broken up in smaller floes, ice type identification gets less reliable (surface scattering can dominate the volume scattering used for MYI type identification). Uncertainty estimates should be mentioned here and also more critically discussed later in the paper (e.g. in 3.1) when the ice types are analysed.

Thank you for pointing this out. The quantification of ice type errors is difficult because it might vary temporally and regionally, also depending on external factors. We have added paragraphs/ sentences in the relevant sections (2.3, 4.4) to discuss the uncertainties of the ice type products more carefully.

p5, l13: if you average them you wouldn't get the monthly displacement but the mean 48h displacement.

We agree that this formulation was misleading, and we have changed it.

p5, l19: hm, the gates are not aligned with the grid. The gates then would be not smooth lines like in the figure but step-wise functions, right? I think that makes flux calculations unnecessary complicated (see below). p5, eq. 2: why is l kept constant at 25km? Depending on the direction of the meridional or zonal component l can increase to sqrt(2)*l = 1.4*l=35km at 45◦ or not?

We calculate the export at the meridional (zonal) gate considering the line section crossing each grid cell. The length of the grid cell is thus a function of the longitude, which is considered in Eq. 2 as mentioned above. We have now changed the formulation and introduced l_uv = l/cos(lambda) to clarify this point.

p5, eq. 3: again, l should not be constant. With 40% the changes in effective length of your grid cell, which you did not take into account, could well dominate your error.

l/cos(lambda) does account for the changing length of the grid cell as a function of longitude. See above.

p6,l1 and ff: Do you apply the error functions from that paper to your datasets or do you use their values? Do these error values then vary in space and time? Shouldn't the error function depend on

the drift dataset or dis Sumata evaluate all your three drift datasets? Please provide some more information and clearer description of method used here.

We do apply the error function from their paper. They provide tables with error estimates of drift in x and y directions for different categories of ice concentration and ice drift speed for all three data sets. We have added a more detailed description of how we retrieve the errors.

p6, Figure 1: make figure larger (full page), arrows are hard to see.

Done.

p7, Figure 2: make larger

Done.

p8, Figure 3: make larger

Done.

p9, l8/9: Do you mean Table 3?

Yes, indeed. Thank you. Corrected.

p9, l18: this is definitely related with the EGC location, which floes along the shelf edge (haven't checked but which probably is at 6W at that latitude then). See e.g. papers by de Steur et al.

We have checked this, and included another reference by Steur et al..

p10, l5: SIC is not shown in Figs 2 & 3. What do you mean?

Thank you for pointing this out. This was indeed not entirely clear from the text. We have now included a figure also showing the ice concentration at the gate for the entire time series to clarify what is meant here.

p10, Table 3: Explain what % MYI means (maybe better in the text). Is a grid cell MYI if there was any MYI detected within the month or does it have to be >50%? What are the uncertainties of this MYI % values? Also, how does the MYI product define the MYI ice type? Is >50% MYI fraction within a grid cell considered MYI or 100% etc.?

The OSISAF ice type product only provides binary values for ice type, e.g. FYI or MYI. The percentage gives the fraction of grid cells that indicate MYI along the gate. We have added a paragraph in section 2.3 to shortly explain how the product is derived and what the uncertainties are.

p11, l4: here you are reporting on per grid cell values. That should be mentioned. Because without the grid size of 25km these values are quite hard to set into relation for the common reader.

Thank you for pointing this out. We have added: „The maximum values have to be considered in relation to the 25 km grid resolution and are likely different on smaller scales".

p11, l10: see comment for Table 3. Explain better what "majority fraction MYI" means.

We added a paragraph in section 2.3 for better explanation.

p11, l19. What are the correlation coefficients? Actually, the exports correspond less from what I would have expected from Fig. 6a. Please explain in more detail why, e.g., NSIDC in 11/12 goes down while the other two go up.

Figure 6a only shows the export along the gate and reveals a strong correlation between the products. We have included monthly export estimates using the different drift estimates. The correlation coefficients between the monthly estimates of the different products are > 0.9 and indicate similar variability. However, it seems that there are seasonal biases, especially considering NSIDC in 2011/2012.

p12, l6: I am not sure I agree with that conclusion. If NSIDC goes down in 11/12 while the other two go up, the variability is quite different, or? Similar for 13/14 to 14/15, where OSISAF & NSIDC show strong increase and IFREMER is more neutral.

As mentioned above, this is mostly due to seasonal differences. Considering the monthly export rates, the correlation coefficients between the products are > 0.9.

p12, l7: It is understood that you do not make a ice drift dataset validation study. However, your export estimates do not agree within their uncertainty estimates. Which means either your uncertainty estimates are wrong or you need to justify why you trust one dataset more than another. To me also the inter-annual variability in Fig. 6b is quite different for the three products. Some explanation for that should be added.

The estimated drift uncertainties do not contain potential biases. In Sumata et al. (2014, 2015), it is shown that these drift products are subject of systematic errors. The reason why we use OSISAF here as a reference is that it shows the best performance in the Fram Strait among the products used in this study. On the other hand, it is shown that the three products coincide quite well regarding the monthly variability (correlation between the products > 0.9 ). Therefore, the main results of our study are independent of the used drift product.

p12, l13: in

Corrected.

p13, Figure 7: shades of gray are hard to discern; caption: remove (b) at the end.

Fixed. We have chosen different line colors in the plot to increase readability.

p14, l1: Not "on the other hand". This argument also supports moving the gate north.

Yes, this word choice might be confusing. Corrected by using "in addition" instead.

p14, l19/20: this is supported by the overlap of STD of this study with previous studies for most months.

Yes, exactly. Exceptions are March and April. But in addition, also the other factors (2-5) will probably play a role here.

p14, l22/23: are there estimates of ice thickness gradient between 80 and 82∘N? What gradient does CS2 show? Can you estimate the thickness gradient to support this argument?

This definitely deserves more research. However, using CryoSat-2 thickness estimates for this purpose is not straight forward. South of 82°N, thickness estimates become more and more uncertain, due to the lower CryoSat-2 orbit coverage and higher ice drift.

[Figure]

*Figure R2: Monthly ASCAT backscatter maps from Ifremer and monthly averaged sea ice type from OSISAF for November 2016 and March 2017. MYI area shows a gain of more than 50% from November to March 2017, which seems unlikely.*

p15, l15: Have also a look at Kwok et al. (2013), which analyses ice area export in connection to different atmospheric indices (AO, DA).

Thanks for pointing on this study. We have cited it with regard to the AO and its linkage to ice export.

p17, l5: what happened to the 7th year 16/17?

This winter season 2016/2017 has been excluded, because some obviously erroneous MYI classification has been observed, see Figure R2. We have been in contact with OSISAF therefore, and they are aware of this behavior. It might be linked to the unusual warm winter in 2016/2017 and needs further investigation. We have added an explanation in the text.

p17, l6: "Scattergram"

We have removed this sentence to reflect changes to the figure.

p17 l10: See my mayor comment at the top. I don't think this is correct. $dV\_MYI/dt$ should almost never be positive.

As stated above, we think that thermodynamic growth can play a role also for MYI, certainly if it is undeformed second-year ice with a decreased thickness after the summer melt. But surely, the uncertainty in the ice type discrimination plays an important role, too. Therefore, we reformulated this paragraph.

p17, l15. hm, that is maybe correct. The word "variations" is not very well defined, maybe better standard deviation? However, only 29% of the variance of $dV\_MYI/dt$ is explained by $Q\_MYI$.

We have already calculated $R^2$. Therefore, it gives us the percentage variation in $dV\_MYI/dt$ explained by $Q\_Ex\_MYI$.

p17, 26: yes, there are some similarities in their variability but actually their variability differs quite significantly and they do not agree within their uncertainty estimates. I find this conclusion too positive or at least need some explanation of the problems. Having results that do not agree within their uncertainty but not to mention that I do not find acceptable. Actually, I would prefer that you guide the reader which estimate they should use or you have to increase the uncertainty estimates.

As mentioned above, the uncertainty estimates do not include a bias correction and the main difference between the products is due to systematic differences. However, the correlation between the monthly volume export derived with the 3 different drift products is > 0.9. We have included the correlation coefficients between the products in the paper now. In the beginning of section 3, we refer to Sumata et al. (2014), which shows that OSISAF ice drift reveals the best performance in the Fram Strait.

p18, l7: "explained" to what degree? Give numbers.

We now state the correlation coefficients.

p18, l9: How is "variability" defined if quantitative numbers are given here?

We acknowledge that using „variability" in this context is a bit confusing. It should be Arctic MYI volume change, since we consider dV/dt.

p18, l11-13: I do not agree with this point. See explanations above.

As mentioned above, we agree that a potential bias due to erroneous ice type classification could affect dV_MYI/dt. We have removed this conclusion from the list.

p19,l17: please provide information how and where to obtain this user guide.

We have added a link that directly points on the pdf. Thank you for pointing on this.

FIxed.

P3L9: "Table 1" -> "Table 3"

Fixed.

[revised manuscript text omitted]

---

## Author Response (AR2)

**Response to 2nd review of the manuscript "Satellite-derived sea ice export and its impact on Arctic ice mass balance" by Ricker et al.**

The manuscript improved from the last version and I now find it suitable for publication in The Cryosphere after some minor points are addressed.

I had three mayor concerns.
The first about the ice volume flux calculation got resolved in the new version. I only have a minor comment left (see below).

My second concern was that the three ice volume export estimates presented here do not agree within their uncertainty estimates. The authors did not change that point but they now more clearly state that there are biases between the datasets, which are not taken into account here.

My third point was was only partially addressed: it was questioned if the residual between MYI export and MYI volume can really be attributed to MYI growth or mainly is an effect of the coarse, binary MYI/FYI classification used in this study. I would have hoped for a more critical discussion of the uncertainties caused by the MYI mask used here. Most areas in the Arctic are a mixture of MYI and FYI. The authors now added a sentence acknowledging the new ice growth in leads but make no efforts to quantify or at least estimate the effect of ice volume growth in leads to the overall mass gain in MYI areas. For example, ice growth of thin ice in winter easily is a magnitude larger than ice growth for MYI and therefore already small percentages of lead area can significantly contribute to the ice volume change within a 25km grid cell, which still would be classified as MYI here. Anyway, we can agree to disagree. In the end it's their conclusion and not mine and the methodology used is explained correctly and this point is not as prominent as before in the conclusions now. As said before, the most basic information was now added.

Thank you for your thorough comments. We agree that there is more work to be done to constrain the uncertainties of the ice type mask and of ice type products in general. Yet, quantifying those errors is a challenging task. Leads are very dynamic, and it is difficult to estimate the area/volume of newly formed ice within leads. Also, strong temperature changes, such as those observed over the last winter, might have an impact. In order to quantify these errors properly, it would require dedicated thorough investigation, which is beyond the scope of the present study, but we do believe that it is important to note here that potential errors/biases exist, and thus we comment on that topic in the manuscript. However, we would also want to stress that this concern does not affect the main conclusion of this paper.

I have a few minor comments left which should be addressed before publication.
Pages and lines refer to the new version of the manuscript with changes marked.

p1,l5: ice drift
Fixed.

p2,l2: what do you mean with "this period"? Oct-Apr? Please clarify. Then this would be inline with my previous criticism that MYI volume change is comparably small.

Yes, Oct-Apr is meant here. Indeed, MYI volume changes are comparably small compared to FYI volume changes due to heavily reduced ice growth. Nevertheless, variations in MYI volume still exist and this is what we try to quantify here. For clarification, we have added the period in the sentence: *"While first-year sea ice (FYI) volume reveals a distinct seasonal cycle between October and April due to thermodynamic growth and new forming ice, multiyear sea ice (MYI) volume shows much smaller changes within the October-April period."*

p2,l14-31: I guess somewhere in this paragraph also the FS ice volume export time series obtained from ULS should be mentioned (Vinje et al., Kwok et al.).

We have slightly altered a few sentences in this paragraph, in order to refer more explicitly to the FS ice volume export time series obtained from ULS (Vinje et al., Kwok et al.).

p4,l7: "IFREMER" according to your own definition

Thanks, fixed.

p5,l4-5: Here one could cite one of the OIB studies supporting this 50% over FYI assumption, e.g., Kurtz et al., 2011 or Newman et al., 2014

We now cite Kurtz et al., 2011.

p5, l16: The OSI-SAF ice type is the same product used for the CS2 thickness retrieval. What for are you using it here in addition? Why are two different user manuals cited (Aaboe et al. and Eastwood et al.). Could this be made consistent? Or are you indeed using two different ice type products?

We use the same product for the CS2 thickness retrieval and for this study, but the version of the user manual has changed (from Eastwood et al. to Aaboe et al.) The reference in 2.2 (AWI CS2 sea ice thickness) is outdated and refers to an older version of the user manual. Thanks for pointing on this inconsistency. The current version is indeed Aaboe et al.. We have therefore replaced Eastwood et al. in section 2.2.

p5,l21: do you mean "new ice forming in openings"? I rather would call them leads

We replaced "openings" by "leads".

p5,l24: "could be a result"

Fixed.

Fig. 1: "Means of Arctic sea ice volume fluxes ..." sounds strange.

We replaced "Means" by "Averages".

p7, eq. 2 & 3: Okay, now I understand the equations. You write "... through the defined gate...", which is correct as the gate only varies between -12° and 20°E. However, maybe it is still worth mentioning that the equations are only valid between -45° and 45°E.

Agreed. We have added this information.

p7,l25: I guess you mean "Zonal uncertainties ..."

Yes, exactly, thanks for pointing out our mistake. We fixed it.

Table 2: Maybe some exceptionally high and low values could be marked and the monthly average for all years could be added to help the reader

We added a column with the mean winter values and highlighted the lowest and highest monthly export.

p13,l6: They also would be larger for larger scales. I would remove that half-sentence. It is enough to remind the reader that these numbers are for a 25km scale.

Agreed. We removed the part of the sentence.

p16,l16: "Sea ice" (I actually do not understand why you have removed the hyphen from all sea-ice volume, sea-ice export etc. word combination. In my view it is correct with hyphen if two expressions like "sea ice" and "volume" are combined).

We think both are acceptable and commonly found in the literature. We have decided to not use a hyphen, but we will leave that decision to the journal copy-editing and typesetting.

[revised manuscript text omitted]